# Uptake of intrauterine contraception after medical management of first trimester incomplete abortion: A cross-sectional study in central Uganda

Herbert Kayiga[1]*, Emelie Looft-Trägårdh[2], Amanda Cleeve[2,3], Othman Kakaire[1], Nazarius Mbona Tumwesigye[4], Musa Sekikubo[1], Joseph Rujumba[5], Kristina Gemzell-Danielsson[2], Josaphat Byamugisha[1]

1 Department of Obstetrics and Gynecology, Makerere University, College of Health Sciences, Kampala, Uganda, 2 Department of Women's and Children's Health, Karolinska Institutet, and WHO collaborating centre, Karolinska University Hospital, Stockholm, Sweden, 3 Department of Global Public Health, Karolinska Institutet, Sweden, 4 Makerere University, School of Public Health, Makerere University, College of Health Sciences, Kampala, Uganda, 5 Department of Paediatrics and Child Health, Makerere University, College of Health Sciences, Kampala, Uganda

* hkayiga@gmail.com

## Abstract

### Background

Although intrauterine devices (IUDs) are readily available in Uganda, their utilization remains low, including within post abortion care. The level and factors associated with uptake of post abortion IUDs, are not well documented. We set out to determine the uptake and factors associated with uptake of post abortion IUDs after medical management of first trimester incomplete abortions in central Uganda.

### Methods

Between February 2023 and September 2023, we conducted a cross-sectional study among women aged 15–49 years, who expressed interest in initiating post abortion intrauterine contraception, and were managed with misoprostol for first trimester incomplete abortions at five health facilities in central Uganda. Healthcare providers received extensive training in post abortion contraceptive counselling and service provision. Data from 650 participants were collected using interviewer administered questionnaires. The primary outcome was uptake of post abortion IUDs, defined as the actual insertion of the post abortion IUDs. Data were collected during a post abortion follow up visit. The determinants of post abortion IUD uptake were assessed using the modified poisson regression.

**Data availability statement:** All the necessary data and materials underlying the findings described have been provided as part of this manuscript. In case any more data or materials are needed, they are readily accessible from the corresponding author on request.

**Funding:** This project was supported by funds from The Swedish Research Council, (Grant 2019-04256) in partnership with Makerere University and the MakRif Project. The content is solely the responsibility of the authors and does not necessarily represent the official views of The Swedish Research Council, Makerere University or the MakRif Project. The funder provided support in the form of research expenses, but did not have any additional role in the study design, data collection and analysis, decision to publish, or preparation of the manuscript. Neither the Swedish Research Council nor Makerere University, the funding organizations, is a commercial institution. Both of these institutions are research and education oriented and not commercial organizations.

**Competing interests:** The authors have declared that no competing interests exist.

**Abbreviations:** DMPA, Injectable Depo-Medroxyprogesterone; IUD, Intrauterine Device; LARCs, Long-Acting Reversible Contraceptives; PAC, Post Abortion Care; PAIC, Post Abortion Intrauterine contraception; SOMREC, The School of Medicine Research and Ethics Committee; UNCST, Uganda National Council for Science and Technology; WHO, World Health Organization.

## Results

The prevalence of post abortion IUD uptake among all women assessed was 370/1911 (19.4%; 95%CI 17.7–21.2). The prevalence of IUD acceptors among those who accepted any form of contraceptives was 370/650 (56.9%; 95%CI 53.1–60.7). Among the other available contraceptive choices, 121(18.6%) women opted for injectable Depo-Provera (DMPA), 116(17.9%) women opted for implants, 35(5.4%) women opted for oral contraceptive pills, 5 (0.8%) of the women opted for condoms, and 3 (0.5%) women for periodic abstinence. The median age of the participants was (27±IQR 30, 23) years. The majority of the participants who had had abortions were living with a partner (84.0%). A third of the participants who had had abortions, had 2−3 children. Nearly 60% of IUD users, opted for the copper IUDs. The post abortion IUD uptake was independently associated with religion- being a Pentecostal (Adjusted PR=2.49, 95%CI= (1.19–5.23), p-value=0.016), monthly earning > one million Ugx (270 USD) (Adjusted PR=1.88, 95%CI= (1.44–2.46), p-value<0.001), and staying <5 kilometres from the health facility (Adjusted PR=1.34, 95%CI= (1.04–1.72), p-value=0.023). Women who were not cohabiting with their partners, were less likely to choose IUDs (Adjusted PR=0.59, 95% CI= (0.44–0.79), p-value=0.001).

## Conclusion

The uptake of IUDs among post abortion women was nearly 60% emphasizing the potential impacts of integrated contraceptive services in Post abortion care. The impact of comprehensive and updated training on post abortion contraceptive counselling, is vital on the uptake of IUDs. Regardless of sociodemographic status, women seeking post abortion care in Uganda should be provided with high-quality integrated services by trained providers offering a range of contraceptive methods. Such efforts may not only prevent unintended pregnancies but also improve health equity across the country.

## Introduction

Unwanted pregnancies end up in induced abortions [1] that may result into unsafe abortions and subsequent abortion-related complications [2]such as sepsis, hemorrhage, infertility and adverse pregnancy outcomes in the subsequent pregnancies [3]. Unsafe abortions in sub-Saharan Africa, have been implicated in 9.9% of the maternal deaths [4]. On the African continent, 460 deaths per 100,000 as compared to 30 deaths per 100,000 in high income countries follow unsafe abortions [5]. A high unmet need for effective contraception in Africa, has been identified as the main driver for the unwanted pregnancies [6].

Uganda has a high total fertility rate of 5.2 births per women [7], and a largely young population where nearly 80% of the population is below 30 years. Nearly 50% of the total population are adolescent girls. The unmet need for modern contraception in Uganda of 27.7%, [8] is higher than the 24.6% reported for the East African region

[9]. As a result, 56% (1.2 million) of all pregnancies in this region are unintended, with 25% of the pregnancies (39 per 1000) ending in abortion [10]. With the abortion laws being restrictive, many Ugandan women experience induced abortions, some of whom end up as unsafe abortions or as maternal deaths [11].

First trimester incomplete abortions are managed either surgically using manual vacuum aspiration or medically using misoprostol [12,13]. Research shows that misoprostol is highly acceptable, effective and safe to use in the management of first trimester incomplete abortion [12]. After surgical management, most of the contraceptive methods including IUDs can be initiated immediately thereafter [14]. In many settings, a high proportion of women never return for in-person follow up visits [15,16]. With fertility returning as early as two weeks of the uterine evacuation [17], there is a need for women to have access to effective contraception to prevent future unwanted and mistimed pregnancies [18].

Although, intrauterine devices (IUDs) are readily available in Uganda, their user rate has stalled at 2% among the currently married and 15% among the sexually active unmarried women [7]. In other parts of Africa, similar trends have been reported [19,20] with IUD user rates of less than 5%. Prior studies in sub-Saharan Africa have associated low user rates with low educational status [8,21], desire for more children [22], myths and misconceptions [23], unsupportive spouses [24], low socio-economic status [25], religion [8,24,26], poor prior pregnancy outcomes [26], inadequate skills among healthcare providers [27] and knowledge gaps about IUDs [18].

Information on rates of IUD uptake within post abortion care, and the factors that are associated with uptake, are lacking in Uganda. Such information may be useful in order to improve post abortion IUD uptake and prevent unintended pregnancies in Uganda. To this end, this study set out to determine the level and factors associated with uptake of post abortion intrauterine contraception following medical management of first trimester incomplete abortions in central Uganda.

## Materials and methods

### Study design

We used a cross-sectional study to determine the level and factors associated with uptake of post abortion IUDs following medical management of first trimester incomplete abortions at five public facilities between 1st February 2023 and 30th September 2023. The participants selected were part of those who were enrolled in the primary study which was non-inferiority open-label randomized controlled study, that compared the expulsion and continuation rates following early insertion (within one week) versus standard insertion (2–4 weeks) after medical management of first trimester incomplete abortion [28]. The trial was registered at ClinicalTrials.gov NCT05343546

### Study setting

Five public health facilities including Kawempe National Referral Hospital, Kayunga Regional Referral Hospital, Mityana General Hospital, Kiganda Health centre IV, and Bukuya Health centre III, were selected from central Uganda. The central region was selected due to the high abortion rate compared to the national average (62 vs. 39 per 1,000 live births) and the large case load of women treated for abortion complications in this setting [10,29,30]. All five health facilities offered free medical and contraception services seven-days a week (24/7) and were at the time of the study, the biggest family planning service providers in central Uganda.

All study sites were located within 1–2 hours from Kampala, the capital city of Uganda. The selected public study sites were equipped to provide comprehensive emergency gynaecological services in rural, peri-urban and urban areas. Routinely at the health facilities, women presenting with symptoms and signs of incomplete abortion are triaged at the emergency wards or the outpatient department. History taking and physical examination are conducted by the attending healthcare providers to confirm the diagnosis of incomplete abortion. Pregnancy dating is done by calculating the weeks of gestation from the first day of the last menstrual period or by ultrasound. Laboratory examinations are usually undertaken if the patient is found to be anemic or has other co-morbidities like febrile illnesses. Patients with first trimester

incomplete abortion based on their clinical status, are managed either medically using Misoprostol or surgically using manual vacuum aspiration. (S1 Appendix) for the descriptive characteristics of the five study sites.

### Participant recruitment

The criteria for identifying study participants with incomplete abortion was based on experience of any of the following conditions: a confirmation of pregnancy by any of the following methods; a positive urine HCG, or calculation from the first day of the last menstrual period. Furthermore, history of lower abdominal pain, and vaginal bleeding, before 12 weeks of gestation was needed to confirm occurrence of first trimester incomplete abortion. Clinical evaluation that included ascertaining cervical dilatation, feeling or visualization of products of conception on vaginal or speculum examination by the clinical team, was conducted to confirm the diagnosis of first trimester incomplete abortion. As indicated in evidence [31], ultrasonography in our study was only used when the clinical team was suspecting incomplete abortion or IUD expulsions.

The research assistants being part of the care teams at the different health facilities, identified and approached potential participants willing to undertake medical management for first trimester incomplete abortion from the emergency gynaecology units. Individuals interested in using post abortion intrauterine contraception, were given all the required information on all available contraceptive methods to enable them make an informed voluntary choice of a family planning method. They were then assessed for eligibility, provided written and oral information about the study. Participants who consented to participate, were enrolled into the study.

### Inclusion criteria

Women 15 years or older having undergone medical management for first trimester incomplete abortion and expressed interest in initiating post abortion intrauterine contraceptive method, within the past four weeks and who were willing to participate in the study, were recruited.

### Exclusion criteria

Participants who were too sick to participate in the study or with confirmed uterine anomalies like bicornuate uterus, ectopic pregnancies, coagulopathies, severe anemia, allergy to copper or levonorgestrel IUDs, or sepsis, were excluded.

### Data collection procedure

All eligible women with first trimester incomplete abortion who opted for medical management of abortion, meeting the inclusion criteria and the WHO medical eligibility criteria for post abortion intrauterine contraception, were invited to participate in the study. Women chose between the copper IUDs (Nova T®, Bayer AG, Berlin, Germany) and the levonorgestrel (LNG) IUDs (Mirena®, Bayer AG, Berlin, Germany). Both copper IUDs and LNG IUDs are highly effective in preventing clinical pregnancies. In the first year of use, copper IUDs have a failure rate of 0.7 pregnancies per 100 women [32] while LNG IUDs are reported to have about 0.1 pregnancies per 100 women [33].

At the post abortion follow-up visit, potential participants were given comprehensive information on either surgical evacuation (uterine aspiration) or medical evacuation with sublingual misoprostol 400 mcg by trained nurse-midwives. Contraceptive counselling on all available contraceptive methods and information about the study was given to all potential participants. We used purposive sampling where participants who met the eligibility criteria and were interested in participating in the study, were enrolled consecutively and given their preferred contraceptive choices, until the desired sample size was achieved. A pretested standard questionnaire was administered by the trained staff on participants' sociodemographic characteristics, contraceptive choices and predictors of the post abortion intrauterine contraceptive uptake.

## Sample size calculation

We used a study that assessed *Post abortion contraceptive uptake among young women in ten countries in South Asia and Africa*, to calculate the sample size for the primary outcome. In this study, the overall uptake of IUDs was 11% [34]. After computation into OpenEpi info sample size calculator, we required 190 participants to answer the objective on uptake of the post abortion IUDs.

## Sample size calculation for factors associated with Post Abortion Intrauterine Contraception after first Trimester Medical PAC (secondary outcomes)

In a study by Makenzius [35], the Age group of 21–25 (OR: 2.35; p < 0.029) was independently associated with contraceptive uptake in PAC. Using OpenEpi info sample size calculator, with OR of 2.35, power of 80%, two-sided confidence level of 95% and level of significance of <0.05, a sample size for associated factors of 642 participants was required. To compensate for any non-response, a total sample of 650 participants was used to determine both primary and secondary objectives for the study. The primary outcome and secondary outcomes (choice of other contraceptive methods) were measured during the post abortion follow-up visit.

**Primary outcome**: Uptake of post abortion intrauterine contraception, was defined as the actual placement of the post abortion IUDs within four weeks of medical management of first trimester incomplete abortion.

**Predictor variables**: Sociodemographic characteristics like age, parity, socio-economic status, education background, health facility factors like availability of the contraceptive mix, were assessed in relation to the uptake of the post abortion IUDs.

## Quality control

**Staff training and recruitment.** Nurse-midwives familiar with the local hospital settings were selected and trained for three days. The training included how to identify potential participants, study procedures and participant recruitment while observing the research ethics in accordance to the Declaration of Helsinki [36]. Thirty research assistants were then selected to collect the data at five study sites over the study period. The nurse-midwives were also trained on how to identify emergencies like severe hemorrhage following the abortion, need for blood transfusion, genital infections like septic abortion and the procedures to undertake so as to inform the obstetric team on duty so that the affected participants could obtain timely emergency care to save their lives.

A pilot study was carried out to pretest and modify the data collection tools. Completed data collection tools were checked daily for completeness and thereafter edited, coded and entered by two data clerks on same day of collection for the five health facilities into KoboToolbox, an online platform for data collection. Data was backed up on external hard drives, encrypted and kept on a secure cloud storage. The database used was password protected and the participants' records were kept under restricted access in a lockable cabin. The research materials were only accessed by the research team in order to protect patient confidentiality and privacy.

## Data management

Electronic questionnaires were designed in KoboToolbox (https://www.kobotoolbox.org/) with programmed logic checks and validation rules. The KoboToolbox software was installed on two tablets for the two data clerks to ensure double data entry. Preset validation checks in KoboToolbox enabled us to ensure that the questionnaires were fully and correctly filled before the data was sunk into the local database. We performed double data entry to ensure consistency between the two entered data. Data was then downloaded into the PAIC database and cleaned using STATA version 15.0 software (Stata Corporation, College Station, TX, USA). Queries were generated and addressed by the principal investigator and research team before final data freeze and analysis.

## Ethical consideration

Ethical approvals were obtained from The Makerere University School of Medicine Research and Ethics Committee, (Mak-SOMREC-2021–131), and Uganda National Council for Science and Technology (HS2111ES). Administrative clearances were obtained from the administrators of the five health facilities. Written informed consent was obtained from all study participants prior to data collection. Participants were reassured that participating in the study was voluntary and that they could opt out of the study without any interference with the relationship with the research team and service delivery. Participants were compensated for their time. Confidentiality and participants' rights were observed throughout the study.

## Analysis

Univariate analysis was done using frequencies and percentages for categorical variables and median and interquartile ranges for continuous variables where appropriate. A test of normality was performed to determine whether data was normally distributed or not. We then compared the background characteristics of the participants who had insertion of the post abortion IUDs to those who opted for other contraceptive methods. Categorical variables were analysed using a Chi-square while continuous variables were analysed using Student's t test for normally distributed data and Wilcoxon test for skewed data.

Bivariate analysis was done using modified poisson regression due to the prevalence being greater than 15% as the best option for cross sectional studies when the outcome of interest is not rare [37]. Though binary logistic regression was the other alternative, it tends to over-estimate the risk when the primary outcome is not rare [38,39]. Crude Prevalent Ratios were obtained and variables with a p-value of less than 0.2, were included into the final multivariable model. To determine factors independently associated with uptake of intrauterine contraception, adjusted Prevalent Ratios were obtained using the modified poisson regression model with the backward elimination model for all variables until the stopping criterion was met.

All statistical tests were two-tailed and variables with two-tailed p-values of less than 0.05 were considered statistically significant. The level of uptake of the IUDs was determined by the computing the proportion of women who had post abortion IUDs inserted as their contraceptive method, out of the women who receive family planning counselling following medical management for first trimester incomplete abortion, expressed as a percentage.

We assessed for confounding by considering a ten percent change in the Prevalence Ratio of a model with the dependent variable and one without. Interaction between the study variables was assessed by forming two-way interaction terms and comparing the models using the likelihood ratio test. Having reviewed literature on the post abortion contraception uptake, variables such as religion, cultural influences and family size [40], participants' prior experiences of contraception, future fertility intentions and desire to meet future reproductive needs [41], that were theoretically plausible to have interaction, were identified especially when they had significant main effect and correlated to each other. We explored their bivariate relationship using correlation coefficients to understand their interaction patterns. Considering the hierarchical principle, we included all variables that had main effects to the IUD uptake. We used a full model that ensured that all variables with potential interaction terms and main effects to the uptake were included. We then used Likelihood Ratio tests for the model comparison. We evaluated for model fit and parsimony to avoid overfitting. We were thereafter able to report the variables we used in the two-way interaction tests when comparing models while using the likelihood ratio tests.

Using the Modified Hosmer-Lemeshow test as the ideal test for goodness of model fit for the modified poisson regression, we obtained P-value=0.57 which is greater than 0.05. This shows that the model's predicted probabilities were well-aligned with the observed outcomes, indicating good model calibration [42].

## Results

### Baseline characteristics of study participants

Between 1st February 2023 and 30th September 2023, we screened 1,911 potential participants who were deemed fit for medical management of first trimester incomplete abortions at the five health facilities. Out of these potential participants,

we enrolled 650 participants who were willing and consented to use post abortion intrauterine contraception within four weeks after medical management of first trimester incomplete abortion. Of the 918 women who met the inclusion criteria and willing to participate in the study, 650/918 (70.8%) had medical management while 251/918 (27.3%) participants had surgical evacuation for their first trimester incomplete abortions, and 17/918(1.9%) participants had exploratory laparotomies following ectopic pregnancies (Fig 1).

The prevalence of post abortion IUD uptake among all women assessed was 370/1911 (19.4%; 95%CI: 17.7–21.2). The prevalence of IUD acceptors among women who accepted any form of contraceptives was 370/650 (56.9%; 95%CI: 53.1–60.7).

Nearly 60% of women who opted for intrauterine contraception, chose the copper IUDs 213(57.6%) while 157(42.4%) women opted for levonorgestrel IUDs. Of the other available contraceptive methods, 121(18.6%) women opted for injectable Depo-Provera (DMPA), 116(17.9%) women opted for implants, 35(5.4%) women opted for oral contraceptive pills, 5 (0.8%) of the women opted for condoms, and 3(0.5%) women opted for periodic abstinence.

A third of the participants were aged between [25–29] years. Over 50% of the participants who had abortions, had had a livebirth before. The majority of the participants who had had abortions were living with a partner (84.0%). Forty percent

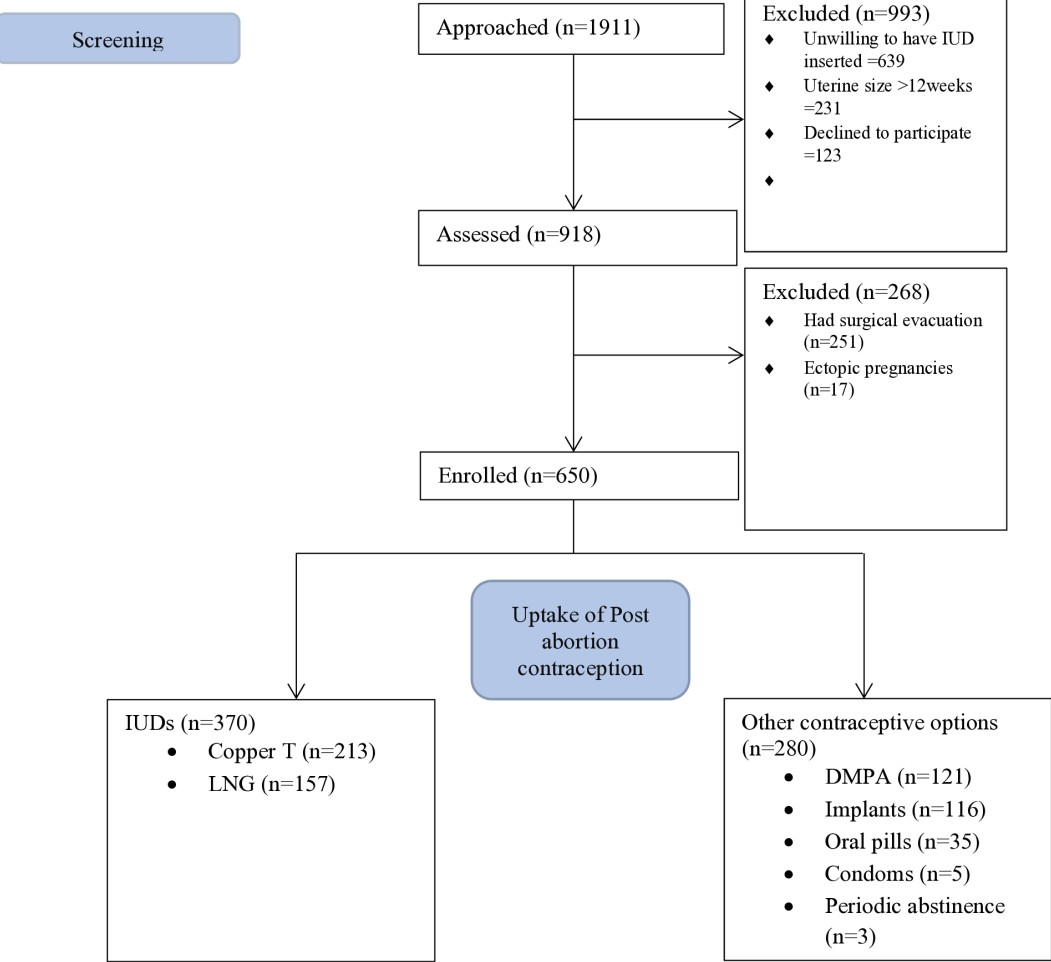

**Fig 1. PRISMA Flow diagram of Participants enrolled in the Uptake of Post abortion intrauterine contraception after medical management of first trimester incomplete abortion. Uptake 2.**

of the participants who had had abortions, had 2–3 children. Only 7(1%) of the participants had a monthly income greater than Ugx shillings 1,000,000/= (USD 270) (Table 1).

**Factors associated with the use of IUDs after the first trimester of incomplete abortion in Uganda**

At bivariate analysis; age, religion, current number of children, occupation, monthly income, distance from health facility, spousal approval of the contraception, and co-habiting status, had p-values less than 0.2 and were utilized for the multi-variable analysis (Table 2).

At multivariable analysis; Women who were Pentecostal or Catholic, earning more than one million Ug shillings (USD 270), staying within five kilometres from the health facilities and living with their partners, were more likely to choose post abortion IUDs.

Religion: Pentecostal women were 2.49 (Adjusted PR = 2.49, 95%CI= (1.19–5.23) p-value = 0.016) times more likely to use IUDs compared to those who were Adventists. Women who were Catholics were 2.13 (Adjusted PR = 2.13, 95%CI= (1.01–4.47), p-value = 0.045) times more likely to use IUDs compared to those who were Adventists.

Socioeconomic status: Women who earned greater than 1,000,000/= (USD 270) were 88% (Adjusted PR = 1.88, 95%CI= (1.44–2.46), p-value<0.001) more likely to use IUDs compared to those who earned less than 50,000/=.

Cohabiting Status: Women who were not living with their partners were 41% (Adjusted PR = 0.59, 95%CI= (0.44–0.79), p-value = 0.001) less likely to use IUDs compared to those who were living with their partners.

Proximity to the health facilities: Women who lived less than five kilometres from the health facility were 34% (Adjusted PR = 1.34, 95%CI= (1.04–1.72), p-value = 0.023) more likely to use IUDs compared to those who lived more than 20 kilometres from the health facility (Table 2).

## Discussion

In this cross-sectional study, we determined the level and factors associated with uptake of post abortion IUDs following medical management of first trimester incomplete abortions at five public facilities in central Uganda. The uptake of post abortion IUDs following medical management of first trimester incomplete abortion was 57%. The factors associated with post abortion IUDs were; religion, woman's income, cohabiting status and participants' distance from their homes to the health facilities.

The uptake of post abortion IUDs in our study is similar to the 60% reported in the Marie Stopes International clinics across Australia [43] but it is significantly higher than the one percent reported at two public hospitals in Kisumu, Kenya. In this Kenyan study, a high proportion of women opted for injectable Depo-Provera (39%) followed by pills (27%) and condoms (25%) [35]. Our study's post abortion IUD uptake is also comparable to the 48% reported in Kalafong Provincial Tertiary Hospital in Atteridgeville, Pretoria, South Africa following uterine evacuation [44]. Although, uptake of post abortion intrauterine contraception has been reported to be higher after surgical as compared to medical evacuation [43,45], the uptake in our study after medical evacuation was higher than the 18.4% reported in the Massachusetts, United States after surgical evacuation [46]. Comparisons of our high uptake rate and rates shown in previous studies must however, be done with some caution, keeping in mind that we included only women who had expressed an interest in initiating an IUD and who attended a follow-up visit in facilities with healthcare providers who had recently received comprehensive training in post abortion family planning counselling and provision. The importance and positive impact of healthcare providers' training on the uptake of post abortion IUDs, has previously been shown by Benson et al. who analyzed client log book data from 921,918 abortion care cases in 4,881 health facilities from July 2011 through June 2015 across ten countries in Asia and sub-Saharan Africa [34]. Trained healthcare providers were more likely to adequately counsel and offer post abortion contraception compared to healthcare providers who were not trained. Hence, acquisition of the appropriate knowledge and skillsets in post abortion intrauterine contraception by healthcare providers in our study could explain the high uptake of the IUDs. Other studies in similar settings have reported that the uptake of contraception was even higher

**Table 1. Sociodemographic and gynaecological characteristics of 650 participants.**

| Variables | Overall (N = 650) (%) | Accepted IUD (n = 370) (%) | Did not accept IUD (n = 280) (%) | Test statistics | P-value |
|---|---|---|---|---|---|
| **Age (years)** | | | | | |
| Median (IQR) | 27 (30,23) | 31 (34,21) | 26 (30,22) | Z (−2.69) | **0.011** |
| <20 | 60(9.3) | 29(4.5) | 31(4.8) | chi² (8.66) | 0.070 |
| 20-24 | 162(24.9) | 80(12.3) | 82(12.6) | | |
| 25-29 | 214(32.9) | 132(20.4) | 82(12.6) | | |
| 30-35 | 153(23.5) | 91 (14) | 62(9.5) | | |
| >35 | 61(9.4) | 38(5.8) | 23(3.5) | | |
| **Religion** | | | | | **0.028** |
| Adventist | 18(2.8) | 5(0.8) | 13(2.0) | chi² (11.24) | |
| Anglican | 170(26.1) | 94(14.5) | 76(11.7) | | |
| Catholic | 210(32.3) | 120(18.5) | 90(13.8) | | |
| Moslem | 113(17.4) | 60(9.2) | 53(8.2) | | |
| Pentecostal | 139(21.4) | 91 (14) | 48(7.3) | | |
| **Prior pregnancy outcome** | | | | | 0.190 |
| Early Neonatal Death (ENND) | 10(1.5) | 6(0.9) | 4(0.6) | Fisher's exact | |
| Intrauterine foetal death (IUFD) | 9(1.4) | 4(0.6) | 5(0.8) | | |
| Induced abortion | 42(6.5) | 20(3.1) | 22(3.4) | | |
| Live birth | 387(59.5) | 232(35.8) | 155(23.8) | | |
| No prior pregnancy | 92(14.2) | 43(6.6) | 49(7.5) | | |
| Spontaneous abortion | 110(16.9) | 65(10.0) | 45(6.9) | | |
| **Number of living children** | | | | | **0.003** |
| 0 | 103(15.8) | 45(6.9) | 58(8.9) | chi² (16.17) | |
| 1 | 137(21.1) | 69(10.6) | 68(10.5) | | |
| 2-3 | 262(40.3) | 160 (24.6) | 102(15.7) | | |
| 4-5 | 115(17.7) | 75(11.5) | 40(6.3) | | |
| >5 | 33(5.1) | 21(3.2) | 12(1.8) | | |
| **Level of education** | | | | | |
| No formal education | 21(3.2) | 11(1.7) | 10(1.5) | Fisher's exact | 0.269 |
| Primary | 197(30.3) | 121(18.6) | 76(11.7) | | |
| Secondary | 368(56.6) | 199(30.7) | 169(26.0) | | |
| Tertiary | 64(9.9) | 39(6.0) | 25(3.8) | | |
| **Occupation** | | | | | 0.176 |
| Employed | 310(47.7) | 185(28.5) | 125(19.2) | chi² (1.83) | |
| Housewife or unemployed | 340(52.6) | 185(28.5) | 155(23.8) | | |
| **Monthly income** | | | | | **0.019** |
| <50,000/= | 307(47.2) | 164(25.2) | 143(22.0) | Fisher's exact | |
| 50,000-499,999/= | 288(44.3) | 169(26.0) | 119(18.3) | | |
| 500,000-999,999/= | 48(7.4) | 31(4.8) | 17(2.6) | | |
| >1,000,000/= | 7(1.1) | 6(0.9) | 1(0.2) | | |
| **Distance from health facility** | | | | | **<0.001** |
| >20km | 76(11.7) | 37(5.7) | 39(6.0) | chi² (19.65) | |
| >10–20 km | 137(21.1) | 66(10.2) | 71(10.9) | | |
| 5–10 km | 167(25.7) | 86(13.2) | 81(12.5) | | |
| <5km | 270(41.5) | 181(27.8) | 89(13.7) | | |
| **Smoking status** | | | | | 0.304 |

*(Continued)*

**Table 1.** (Continued)

| Variables | Overall (N = 650) (%) | Accepted IUD (n = 370) (%) | Did not accept IUD (n = 280) (%) | Test statistics | P-value |
|---|---|---|---|---|---|
| No | 631(97.1) | 357(54.9) | 274(42.2) | Fisher's exact | |
| Yes | 19(2.9) | 13(2.0) | 6(0.9) | | |
| **Marital status** | | | | | **<0.001** |
| Not living with partner | 104(16.0) | 36(5.5) | 68(10.5) | $chi^2$(24.1) | |
| Living with the partner | 546(84.0) | 334(51.4) | 212(32.6) | | |
| **Number of sexual partners** | | | | | 0.37 |
| 1 | 520(80.0) | 301(46.3) | 219(33.7) | $chi^2$ (0.79) | |
| 2-3 | 130(20.0) | 69(10.6) | 61(9.4) | | |
| **Accompanied by partner to the clinic** | | | | | 0.483 |
| No | 469(72.2) | 263(40.5) | 206(31.6) | chi2 (0.492) | |
| Yes | 181(27.8) | 107(16.5) | 74(11.4) | | |
| **HIV Status** | | | | | 0.850 |
| Negative | 621(95.5) | 353(54.4) | 268(41.2) | Fisher's exact | |
| Positive | 29(4.5) | 17(2.6) | 12(1.8) | | |

N stands for absolute frequency and % stands for Row percentage, IQR, Interquartile range

when the available healthcare providers were more skilled in intrauterine contraception, and in the facilities that had procedure rooms for evacuation [41,47,48].

About 42.4% of study participants who took up post abortion IUDs chose the hormonal IUDs. Although levonorgestrel IUDs have been reported to have advantages [49,50] such as reduced menstrual loss, and dysmenorrhea, compared to the copper IUDs, nearly 60% of the participants in the study preferred copper IUDs over levonorgestrel IUDs. Despite the said advantages, a number of African women desire to have periods as an assurance of their fertility and also might fear side effects associated with levonorgestrel IUDs [18,51].

Findings in our study show that religion has an impact on the uptake of post abortion IUDs. The influence of religion on contraceptive use has been highlighted in other studies. For instance, Addai, [52] identified religion as a strong determinant to the uptake of contraception in Ghana. In the United States, Catholics were reported to have low contraceptive user rates [53]. These results are contrasted by Bakibinga 2016, who found that religion didn't have any influence on the uptake of contraception [54]. The observation of more Pentecostal Christians and Catholics taking on post abortion IUDs as reported in our study could suggest a mindset adjustment among the individuals and a collective effort from the religious leaders to advocate for effective contraception like IUDs. In Western Kenya, when religious leaders were offered with the appropriate family planning knowledge and involvement at six faith-based organizations, the uptake of modern contraception increased by tenfold. This underscores the impact of religious leaders' involvement and advocacy in the uptake modern contraception [55].

Our study results underscore the importance of women's socio-economic status on contraceptive uptake. Although contraceptives were free of charge in our study, we found that women who earned more than one million Ugandan shillings (USD 270) were more likely to opt for IUDs as compared to house wives or those who were unemployed. Wealthier women have been reported to utilize the IUDs more than unemployed women [8]. The wealth status is believed to follow women's education status that gives the women more power to decide their contraceptive choices. As women get more years of education, their wealth status improves. This empowerment enables wealthier women to make more informed decision in regard to their contraceptive needs as compared to women of less education and low social economic status [56]. Women of low socioeconomic status in Marie Stopes International clinics across Australia were noted to leave

**Table 2. Regression analysis of the factors associated with the Uptake of IUDs after First Trimester Incomplete Abortion managed with Misoprostol (n = 650).**

| Variables | Other contraceptive uptake (n = 280) | Unadjusted PR (CI 95%) | Adjusted PR (CI 95%) | P-value |
| --- | --- | --- | --- | --- |
| | IUD uptake (n = 370) | | | |
| **Age (years)** | | | | |
| <20 | 31(11.1) | 29(7.8) | 1 | 1 | |
| 20-24 | 82(29.3) | 80(21.6) | 1.00(0.74-1.35) | 0.81(0.59-1.11) | 0.977 |
| 25-29 | 82(29.3) | 132(35.7) | 1.23(0.93-1.62) | 0.80(0.58-1.12) | 0.121 |
| 30-35 | 62(22.1) | 91(24.6) | 1.22(0.92-1.63) | 0.78(0.55-1.10) | 0.114 |
| >35 | 23(8.2) | 38(10.3) | 1.31(0.96-1.80) | 0.87(0.60-1.27) | 0.081 |
| **Religion** | | | | |
| Adventist | 13(4.6) | 5(1.4) | 1 | 1 | |
| Anglican | 76(27.2) | 94(25.4) | 1.99(0.93-4.25) | 2.09(0.99-4.41) | 0.054 |
| Catholic | 90(32.2) | 120(32.4) | 2.06(0.97-4.38) | 2.13(1.01-4.47) | **0.048** |
| Moslem | 53(18.9) | 60(16.2) | 1.91(0.89-4.11) | 1.99(0.94-4.22) | 0.061 |
| Pentecostal | 48(17.1) | 91(24.6) | 2.35(1.10-5.00) | 2.49(1.19-5.23) | **0.015** |
| **Current number of children** | | | | |
| 0 | 58(20.7) | 45(12.2) | 1 | 1 | |
| 1 | 68(24.3) | 69(18.6) | 1.15(0.88-1.52) | 1.02(0.77-1.36) | 0.825 |
| 2-3 | 102(36.4) | 160 (43.2) | 1.40(1.10-1.78) | 1.19(0.91-1.57) | 0.414 |
| 4-5 | 40(14.3) | 75(20.3) | 1.51(1.17-1.95) | 1.28(0.95-1.72) | 0.219 |
| >5 | 12(4.3) | 21(5.7) | 1.46(1.04-2.04) | 1.22(0.84-1.80) | 0.474 |
| **Occupation** | | | | |
| Employed | 125(44.6) | 185(50.0) | 1 | 1 | |
| Housewife or unemployed | 155(55.4) | 185(50.0) | 0.91(0.80-1.04) | 1.03(0.85-1.25) | 0.619 |
| **Monthly income (1 USD/3700 Ugx)** | | | | |
| <50,000/= | 143(51.0) | 164(44.3) | 1 | 1 | |
| 50,000-499,999/= | 119(42.5) | 169(45.7) | 1.12(0.98-1.30) | 1.15(0.95-1.40) | 0.113 |
| 500,000-999,999/= | 17(6.1) | 31(8.4) | 1.21(0.96-1.53) | 1.25(0.93-1.68) | 0.107 |
| >1,000,000/= | 1(0.4) | 6(1.6) | 1.60(1.16-2.21) | 1.88(1.44-2.46) | **< 0.001** |
| **Distance from health facility** | | | | |
| >20km | 39(13.9) | 37(10.0) | 1 | 1 | |
| 16–20 km | 29(10.4) | 26(7.0) | 0.97(0.68-1.40) | 0.91(0.63-1.32) | 0.743 |
| 11–15 km | 42(15.0) | 40(10.8) | 1.00(0.73-1.38) | 0.96(0.70-1.32) | 0.975 |
| 5–10 km | 81(28.9) | 86(23.3) | 1.06(0.80-1.39) | 0.99(0.75-1.32) | 0.974 |
| <5km | 89(31.8) | 181(48.9) | 1.34(1.08-1.76) | 1.34(1.04-1.72) | **0.035** |
| **Cohabiting status** | | | | |
| Living with a partner | 212(75.7) | 334(90.3) | 1 | 1 | |
| Not living with a partner | 68(24.3) | 36(9.7) | 0.57(0.43-0.74) | 0.59(0.44-0.79) | **< 0.001** |

p-value <0.05 were significant.

without their intrauterine contraception of preference [43] as compared to those in the higher economic quintiles thereby being predisposed to subsequent unintended pregnancies. These findings highlight opportunities to counteract reproductive health inequities. Policy makers in Uganda should consider subsidizing contraceptives to ensure access to IUDs for women in lower economic quintiles as well as strengthening efforts towards female education for informed contraceptive decision making.

In our study, women who were co-habiting with their partners were more likely to use post abortion intrauterine contraception than single women. This finding has been reported in other studies [35,57,58]. The higher uptake of post abortion IUDs among women in union could be resulting from influence by their partners [59]. The emotional and physical support from their partners during the management of the abortions could have influenced the higher uptake of post abortion IUDs among the marrieds in our study. Similar findings have been reported by Kayi et.al in Ghana [57]. The uptake could also result from healthcare provider bias with providers being less likely to push LARCs on young single women in sub-Saharan Africa with a notion that most single women might be having multiple sexual partners, which could heighten the risk of PID with IUD use [60,61]. Young and unmarried women are commonly encouraged to abstain for fears of becoming infertile following IUD use [62–64]. In Uganda, 44.5% of the pregnancies are unintended [65]. It's likely that married women have an immediate danger of becoming pregnant shortly after an abortion, hence the increased urgency to adopt a family planning method.

Our study findings suggest that proximity to health facilities influences the uptake of post abortion intrauterine contraception. The fact that after medical management women were expected to return for insertion could be the explanation that women who lived closer to the health facilities were more likely to have their IUDs inserted than those who stayed far. Time that would have been wasted in transit as women travel long distances to seek family planning consultations, is reduced by women staying closer to the health facilities. In similar settings as noted by Achana in Upper East Region of Ghana, women staying within two kilometres were 36% more likely to use contraception than those who lived further [66]. Staying within five kilometres to the health facilities among the rural married women was associated with increased utilization of modern contraceptive methods in Rural Ethiopia [67]. Policy makers ought to facilitate services that ensure quick start of family planning methods can integrate in contraceptive counselling and provision on the first contact after abortion treatment. Such approaches can ensure that women who live far do not have to return for follow up visits. Follow up visits for women staying far could be prioritized to be over phone for a majority of women. Prior studies have shown higher continuation rates and patient satisfaction with early intrauterine contraception within a week of medical management of abortions than with the standard insertion between 2–4 weeks later [68]. There's also no need for back up contraception with early insertion as implemented for standard care of inserting IUDs 2–4 weeks later [69,70].

The strength of this study lies in the fact that our sample size of 650 participants was large and included both rural and urban women giving us the ability to generalize our study findings. Participants were comprehensively counseled on a wide range of contraceptive methods and voluntarily adopted a method of their choice thereby minimizing social desirability bias. The healthcare providers had extensive trainings in post abortion family planning counselling and provision prior to the study. The acquired knowledge and skillsets in post abortion contraception, was a backbone that could have motivated women to fully participate from start to end of the study.

We were however unable to determine the continuation rates among the study participants as the study design was cross-sectional. Interviewer administered questionnaires were used to collect the data and this could have led to social desirability bias in the study. Healthcare providers' and couples' perceived barriers and motivators as factors for the uptake of post abortion contraception, could not be determined in the current study but should be explored further. We conducted our study in public facilities, where family planning services are offered free of charge. The uptake and factors associated with the uptake of post abortion intrauterine contraception could differ in the private facilities where family planning methods are paid for. In our study, only women who expressed interest in initiating intrauterine contraceptive method, and attended the post abortion follow up visits, were included in the study. Our results may not be comparable to other similar studies that look at post abortion uptake in general after medical management of first trimester incomplete abortions.

## Conclusion

The uptake of IUDs among post abortion women was nearly 60% emphasizing the need to integrate IUD services in the post abortion care package. Our study findings accentuate the impact of comprehensive and updated training on post

abortion contraceptive counselling on the uptake of IUDs. Efforts are now needed to ensure that all women seeking post abortion care in Uganda, regardless of sociodemographic status, are provided high-quality integrated services by trained providers, that offer a range of contraceptive methods, including IUDs. Such efforts may not only prevent unintended pregnancies but also improve health equity across the country.

## Supporting information

**S1 Appendix. Description of study sites.**
(DOCX)

## Acknowledgments

We extend our sincere appreciation to the research team and study participants for making this study a reality. We are indebted to the study coordinators (Diane Achanda Genevieve and Diana Nankabirwa), and the Administrators of the health facilities for the support they gave us throughout the study. I am so humbled by the support and guidance accorded to us by Prof Dan Kabonge Kaye, Dr John Mukisa, Ms. Proscovia Nakasujja, the Doctoral committee and my PhD supervisors at Makerere University Kampala Uganda, Karolinska Institutet, and WHO collaborating centre, Karolinska University Hospital, Stockholm, Sweden.

## Author contributions

**Conceptualization:** Herbert Kayiga, Emelie Looft-Trägårdh.

**Data curation:** Herbert Kayiga.

**Formal analysis:** Herbert Kayiga, Othman Kakaire, Nazarius Mbona Tumwesigye, Musa Sekikubo, Joseph Rujumba, Kristina Gemzell-Danielsson, Josaphat Byamugisha.

**Funding acquisition:** Amanda Cleeve, Kristina Gemzell-Danielsson, Josaphat Byamugisha.

**Investigation:** Herbert Kayiga.

**Methodology:** Herbert Kayiga, Emelie Looft-Trägårdh.

**Resources:** Herbert Kayiga, Kristina Gemzell-Danielsson, Josaphat Byamugisha.

**Supervision:** Amanda Cleeve, Othman Kakaire, Nazarius Mbona Tumwesigye, Musa Sekikubo, Joseph Rujumba, Kristina Gemzell-Danielsson, Josaphat Byamugisha.

**Writing – original draft:** Herbert Kayiga.

**Writing – review & editing:** Herbert Kayiga, Emelie Looft-Trägårdh, Amanda Cleeve, Othman Kakaire, Nazarius Mbona Tumwesigye, Musa Sekikubo, Joseph Rujumba, Kristina Gemzell-Danielsson, Josaphat Byamugisha.

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
