## [Decision Letter · Decision Letter 0]

29 Jan 2025

Dear Dr. Kayiga,

Thank you for submitting your manuscript to PLOS ONE. After careful consideration, we feel that it has merit but does not fully meet PLOS ONE’s publication criteria as it currently stands. Therefore, we invite you to submit a revised version of the manuscript that addresses the points raised during the review process.

Since interviewer administered questionnaire was utilized, there may Social desirability bias. Authors may discuss this in the limitation of the study.Authors may need to describe when (in weeks post miscarriage) IUCD will inserted post abortion. What evidence or symptoms will support IUCD insertion and what clinical features will make them defer IUCD insertionWas WHO eligibility criteria utilized in the counselling of participants?Authors may need to to have three groups. Those who accepted IUCD, Those who accepted other contraceptives and those who declined contraceptives Are there reasons for declining contraceptive among the populations such as myths and misconceptions? What are they if any?Please describe the IUD offered, The constituent and its  efficacyLine 230. Please state the country origin and the company producing STATA. Line 247: Fischer's exact test is not used for skewed data. It is used in place of Chi square if the expected frequencies in the cells < 5. (please confirm its use)What non-parametric equivalent was utilized in place of Student's ttest where appropriate? (Please note that "S" in Student's ttest is capital letter as it is a name of a person)Line 249 - 250 : Authors stated "Bivariate analysis was done using modified Poisson regression due to the prevalence being greater than 15%" I am a bit confused about this sentence. Authors may explain further and cite appropriate references.Line 251: Please change "multivariate" to "multivariable".  Please state the type of model building. stepwise forward? backward elimination or prior selections?Line 257. The prevalence should be reported as a percentage and its 95% confidence interval. Please authors should state 2 different prevalences: Prevalence of IUD among all women assessed and prevalence of IUD acceptors among those who accepted any form of contraceptives. (with their 95% confidence interval)line 260 - 261: Authors stated : "Interaction between the study variables was assessed by forming two-way interaction terms and comparing the models using the likelihood ratio test" Which variables were utilized for the interaction tests? why and why not? Is the selection based on literature of knowledge of the field? if based on the literature, please provide evidence/citationsAuthors may need to state if the analysis was two tailed or one-tailed Line 269- 273. Authors stated  "Between 1st February 2023 and 30th September 2023, we screened 1,911 potential participants  who received medical management of first trimester incomplete abortions and were willing to take up post abortion intrauterine contraception at the five health facilities. Out of these 271 potential participants, we enrolled 650 participants. The proportion of women who took up 272 IUDs after medical management of first trimester abortion, was 370/650 (56.9%)"This is not clear. Please authors may need to draw a flow diagram to aid understanding.  How was the 650 participants recruited from 1,911? Do you mean 1911 participants were ready to take any contraceptive or they were ready to take IUD? Becuase you now had a proportion 370/650? Who are the denominator here? Please add 95%CI to the prevalence of IUD uptakeLine 273: Majority generally means percentage of >70%. and not >50%line 280: authors stated "A high proportion of the participants (56.6 %)"  56.6% is "a little above half" and not "high proportion"please show the mean age in the Table 1line 283 -284: authors stated ".......and had 2-3 children 283 (38.5%)." 38.5% is not "majority". it is about "one-third"Table 1 should a column for those that accepted IUD, another column for those that did not take IUD , another column for those that did not accept any contraceptive and then the total. For each variable, you may use appropriate statistical tool to compare the variables across the categories of studied groupsTable 1. Please report the mean or median age (as the case may be) in the TableTable 1:  Add year to age i.e "age (years)Table 1. Change "below 20 years" to "<20"Table 1. Change "Greater than 35years" to ">35" Table 1: "Number of living children" is a discrete variable and may not be normally distributed. Authors should check for normality. If not normally distributed median, IQR should be reported instead of mean, SD.Table 1. SD: Define SD as footnote of the table. Add appropriate symbol in the Table. See PLOS one website on formatting tablesTable 1. Some variables in Table 1 are not sociodemographic characteristics. Some are obstetrics or gynaecological characteristics. Sot the title may be "sociodemographic and gynaecological characteristics"Table 1: HIV status is not part of the sociodemographic characteristicsTable 1 . please expunge "Current uptake of Post abortion contraceptive method(N=650) " from Table 1. You may even produce appropriate chart/figure with this.Table 2. "Unadjusted PR (CI 95%)" shifted from where it should be in the Table. please correct itTable 2. "NB". What is "NB" in the footnote? I will advise you remove itDiscussion: You should briefly restate your aim at the beginning of your discussion.  Line 292 - 295: Authors stated "At bivariate analysis; age, religion, current number of children, occupation, monthly income, 292 distance from health facility, spousal approval of the contraception, and co-habiting status, 293 were statistically associated with the use post abortion IUDs after the first trimester of 294 incomplete abortion with a p-value less than 0.2.(Table 2) " P-value pf 0.2 does not suggest Statistical significance. What is means are that they were variables that were utilized for the multivariable analysis because their P-values were < 0.2. Not because they are significant. Authors should rework this sentence Libe 297: change "multivariate" to "multivariable".  So this model building is with a binary variable of (IUD uptake Vs no IUD uptake)???  this may mean that "no IUD uptake" is either uptake of  other contraceptives or non-use of contraceptives??? Please clarify. The outcome of the model should be a binary variable of either (IUD uptake Vs no contraceptive uptake) or (IUD uptake Vs other contraceptive uptake). Your interpretation of the results should clearly incorporate your outcome out of the two scenarios stated aboveLine 430: change "couldn’t " to could not. avoid contracted words in scientific writings?

line 479: Availability of data and Materials: Authors may attach as supplementary materials, the anonymized minimal data underlying this study. It may be in excel, or stata format. (Except there are strong ethical and legal reasons against it) 

We look forward to receiving your revised manuscript.

Kind regards,

Gbenga Olorunfemi, MBBS,MSC,FMCOG,FWASC,FWACOG

Academic Editor

PLOS ONE

Journal Requirements:

2. Please include a complete copy of PLOS’ questionnaire on inclusivity in global research in your revised manuscript. Our policy for research in this area aims to improve transparency in the reporting of research performed outside of researchers’ own country or community. The policy applies to researchers who have travelled to a different country to conduct research, research with Indigenous populations or their lands, and research on cultural artefacts. The questionnaire can also be requested at the journal’s discretion for any other submissions, even if these conditions are not met. Please find more information on the policy and a link to download a blank copy of the questionnaire here: https://journals.plos.org/plosone/s/best-practices-in-research-reporting . Please upload a completed version of your questionnaire as Supporting Information when you resubmit your manuscript.

Please confirm at this time whether or not your submission contains all raw data required to replicate the results of your study. Authors must share the “minimal data set” for their submission. PLOS defines the minimal data set to consist of the data required to replicate all study findings reported in the article, as well as related metadata and methods (https://journals.plos.org/plosone/s/data-availability#loc-minimal-data-set-definition ).

If your submission does not contain these data, please either upload them as Supporting Information files or deposit them to a stable, public repository and provide us with the relevant URLs, DOIs, or accession numbers. For a list of recommended repositories, please see https://journals.plos.org/plosone/s/recommended-repositories .

5. Please include captions for your Supporting Information files at the end of your manuscript, and update any in-text citations to match accordingly. Please see our Supporting Information guidelines for more information: http://journals.plos.org/plosone/s/supporting-information .

Reviewers' comments:

Reviewer's Responses to Questions

**Comments to the Author**

1. Is the manuscript technically sound, and do the data support the conclusions?

Reviewer #1: Yes

Reviewer #2: Yes

Reviewer #3: Yes

Reviewer #4: Yes

2. Has the statistical analysis been performed appropriately and rigorously?

Reviewer #1: Yes

Reviewer #2: No

Reviewer #3: Yes

Reviewer #4: Yes

3. Have the authors made all data underlying the findings in their manuscript fully available?

Reviewer #1: Yes

Reviewer #2: Yes

Reviewer #3: Yes

Reviewer #4: Yes

4. Is the manuscript presented in an intelligible fashion and written in standard English?

Reviewer #1: Yes

Reviewer #2: Yes

Reviewer #3: Yes

Reviewer #4: Yes

Reviewer #1: REVIEW OF ARTICLE TITLED Uptake of Intrauterine Contraception after Medical Management of First Trimester Incomplete Abortion: A Cross-sectional study in central Uganda

Thank you for asking me to review the above titled article. My comments are as follows

• The article is well titled with an appreciation of its content and the methodology expected.

• The authors have chosen a contemporary topic aiming to present components of post abortion care as important strategies to prevent morbidity and mortality from unsafe abortion and its sequelae. This should be applauded

• The abstract is well written giving an overview of the aim of the study its methodology results and conclusion

• The introduction presents the background and basis for undertaking the study and by the end the reader appreciates the justification for this research paper

• The study design is appropriate for research of this nature

• There is no segregation of the index abortions into spontaneous or induced in this write up , and if induced what were the reasons for initiating the termination of pregnancy. This may have a bearing on choosing to have contraception after post abortion care or not

• How did the patients arrive at the decision to undergo medical management? Did they request for it ab initio or were they counselled on all options and allowed to choose? Its not clearly indicated here

• Also who did the counselling for options of therapy? Was it done by those providing the definitive care ? and were they part of the staff trained for this study?

• Just out of curiosity, of the 1911 women recruited you enrolled 650 which was your sample size, but why the drop off of over 65% ? were there any specific trends noted or was it just the exclusion criteria?

• The presentation of results are acceptable and they derive from the aim and purported objectives of this research

• The discussion is well detailed and easy to read, they are based on the results presented with logical arguments and submissions

• The conclusion is acceptable

The study is an interesting one and shines a light on a topic that is usually taboo in Africa. The authors have presented a detailed rendition of their work .

I believe if the queries raised are answered appropriately it can be considered for publication

Thank you

Reviewer #2: Most of unwanted pregnancies end up in induced abortion ......68

Unnecessary references 5......70

Statement with ref 7, check and revise......73

Using 7 references for a statement seem overboard, 2-3 max would be advised....90,91

The content of description of the study setting is too much. Summary and make it more concise especially the management protocol of miscarriage.

Appendix should come after the whole work, not within the main text.

Participants recruitment description also too expansive.

Could have been better to use iud uptake prevalence from study in Uganda and not south Asia or another country. .....187

Why using 2 different sample size calculation. Better to stick with the one gotten from Makenzius which emphasized post abortion iud uptake.

The methodology is too voluminous...all the sections

Statement in line 363-364...reference

Revise the statement in line 391.

….in our study as similar finding was report by Kayi et al in Ghana [ 56]

More point high uptake among married than unmarried... young, unmarried low use could be healthcare providers influence as most unmarried, single have multiple sexual partners which could heighten the risk of PID with iucd use

Reviewer #3: Line 136, why do they have to use Abdominopelvic USG to ascertain cervical dilation? Can’t that be assessed through digital pelvic examinations or speculum examinations or by the use of a transvaginal USG?

Do they use ultrasound to confirm that the uterus is empty before inserting the IUDs post miscarriage?

Line 167 .what is the legal age of consent in Uganda? Are teenagers below 16 years allowed to give consent on their own for the procedure and to participate in a research?

Lines 330-331, what proportion of the respondents in your study had medical and surgical termination of their pregnancies?

Lines 350-353, could that not also be that the African Women is afraid of the side effects associated with the hormonal IUDs?

Lines 382-384 contradict line 370 which states that contraceptive services are free. Or are there other public health facilities in Uganda where contraceptive services eg IUDs are charged?

The writers may attend to the comments and the questions above, and update the paper where necessary.

Aside that I find the paper very impressive and should be accepted for publication for the benefit of all.

Thank you.

Reviewer #4: The research presented by the authors is interesting providing critical insights on the factors that informed the uptake of Intrauterine contraceptive device for women who had a medical management of incomplete abortion in Uganda.

The objectives of the study are well defined, and methodology adequately described with an appropriate statistical analytics strategy.

However, my suggestions to the authors to improve the quality of the manuscript are as follows:

1. On line 154, the authors should consider removing measuring the Symphysio Fundal height (SFH) for diagnosing first trimester abortion as the uterus remains a pelvic organ until after the 12th week, and this is often immeasurable by SFH, except by a bimanual examination.

2. On line 302, women who were Catholics were 2.13 times more likely, not 2.15 times more likely.

3. I would suggest the authors consider using Catholics as the reference group as opposed to the Adventist women in the multivariate analysis as this religious group have been historically averse to the use of contraception.

4. Since the cadre of healthcare workers (physician, nurse or community health worker) who delivered the contraceptive counselling was not specified at the methodology stage, adjusting for it in the regression analysis may remove possible confounding. The gap in personnel knowledge may impact on the choice of IUD contraception or otherwise.

5. In table 2, it is important to state there was no difference among different age groups on the uptake of IUD contraception as this contrast with the cited literature by Makenzie et al on line 192 that the age groups of 21-25 was independently associated with contraceptive uptake in post-abortal care.

Overall this is an interesting research but needs addressing of the points raised.

**Do you want your identity to be public for this peer review?** For information about this choice, including consent withdrawal, please see our Privacy Policy

Reviewer #1: No

Reviewer #2: No

Reviewer #3: No

Reviewer #4: **Yes: ** John Omole-Matthew MBBS MWACS MPH

---

## [Author Response · Author response to Decision Letter 1]

13 Mar 2025

COLLEGE OF HEALTH SCIENCES

SCHOOL OF MEDICINE

DEPARTMENT OF OBSTETRICS AND GYNAECOLOGY

13th March 2025

RESPONSE TO REVIEWS COMMENTS:

Object: PONE-D-24-39454 “UPTAKE OF INTRAUTERINE CONTRACEPTION AFTER MEDICAL MANAGEMENT OF FIRST TRIMESTER INCOMPLETE ABORTION: A CROSS-SECTIONAL STUDY IN CENTRAL UGANDA”

With great pleasure, I’m thankful for your comments towards our manuscript. In response to the reviewers’ comments sent to us on 13th March 2025, we have revised the manuscript accordingly.

Comment Response to Comment Page No. and Line

Dear Dr. Kayiga,

We've checked your submission and before we can proceed, we need you to address the following issues:

1. However, we note that there is identifying data in the Supporting Information file 'Use of IUDs updated..dta'. Prior to sharing human research participant data, authors should consult with an ethics committee to ensure data are shared in accordance with participant consent and all applicable local laws. Data sharing should never compromise participant privacy. It is therefore not appropriate to publicly share personally identifiable data on human research participants.

The following are examples of data that should not be shared:

- Name, initials, physical address

- Ages more specific than whole numbers

- Internet protocol (IP) address

- Specific dates (birth dates, death dates, examination dates, etc.)

- Contact information such as phone number or email address

- Location data

- ID numbers that seem specific (long numbers, include initials, titled “Hospital ID”) rather than random (small numbers in numerical order)

Please remove or anonymize all personal information (Study Identification Number, Date of Recruitment, Religion, Register Unit Number, etc.), ensure that the data shared are in accordance with participant consent, and re-upload a fully anonymized data set. We also request that the dataset be shared in Excel (.xlsx or .csv) format. Please note that spreadsheet columns with personal information must be removed and not hidden as all hidden columns will appear in the published file.

In addition, please upload the following files as 'Other' files instead of Supporting Information files:

Full registration PhD-3.pdf

Institutional clearance for all Health Facilities FP.pdf

UNCST approval letter Family planning.pdf

Revised consent forms 2.pdf

Approved protocol-2.pdf

Approval letter SOMREC.pdf

Approval letter.pdf

We've returned your manuscript to your account. Please resolve these issues and resubmit your manuscript within 21 days. If you need more time, please email the journal office at plosone@plos.org. We are happy to grant extensions of up to one month past this due date. If we do not hear from you within 21 days, we will withdraw your manuscript.

t.

Identifying data has been removed from the shared stata dataset as advised by the editor.

Identifying data has been removed from the shared stata dataset as advised by the editor. An Excel format has been used as advised.

Spreadsheet columns with personal information have been removed as advised.

Other files have been used for the mentioned files as advised.

Thanks so much for quick response otherwise.

Yours truly,

Dr Herbert Kayiga

Lecturer/Corresponding Author,

Department of Obstetrics and Gynaecology

Makerere University College of Health Sciences

+256777855063, hkayiga@gmail.com

---

## [Editor Report · Decision Letter 1]

22 Apr 2025

Dear Dr. Kayiga,

Thank you for submitting your manuscript to PLOS ONE. After careful consideration, we feel that it has merit but does not fully meet PLOS ONE’s publication criteria as it currently stands. Therefore, we invite you to submit a revised version of the manuscript that addresses the points raised during the review process.

**Authors did not attach a point-by-point rebuttal addressing the comments of the reviewers. For ease of reference, authors are to attach a rebuttal addressing all the comments of the reviewers and the editorial comments.  A number line is also necessary for ease of reference. When the point-by-point rebuttal and a revised manuscript showing the marked reviews is available, the manuscript will be sent to the reviewers for review and comments**

We look forward to receiving your revised manuscript.

Kind regards,

Gbenga Olorunfemi, MBBS,MSC,FMCOG,FWASC,FWACOG

Academic Editor

PLOS ONE

---

## [Author Response · Author response to Decision Letter 2]

23 Apr 2025

COLLEGE OF HEALTH SCIENCES

SCHOOL OF MEDICINE

DEPARTMENT OF OBSTETRICS AND GYNAECOLOGY

26th February 2025

RESPONSE TO REVIEWS COMMENTS:

Object: PONE-D-24-39454 “UPTAKE OF INTRAUTERINE CONTRACEPTION AFTER MEDICAL MANAGEMENT OF FIRST TRIMESTER INCOMPLETE ABORTION: A CROSS-SECTIONAL STUDY IN CENTRAL UGANDA”

With great pleasure, I’m thankful for your comments towards our manuscript. In response to the reviewers’ comments sent to us on 30th January 2025, we have revised the manuscript accordingly.

Comment Response to Comment Page No. and Line

Journal Requirements:

Editorial Comments

Since interviewer administered questionnaire was utilized, there may Social desirability bias. Authors may discuss this in the limitation of the study.

Authors may need to describe when (in weeks post miscarriage) IUCD will inserted post abortion.

What evidence or symptoms will support IUCD insertion and what clinical features will make them defer IUCD insertion?

Was WHO eligibility criteria utilized in the counselling of participants?

Authors may need to to have three groups. Those who accepted IUCD, Those who accepted other contraceptives and those who declined contraceptives

Are there reasons for declining contraceptive among the populations such as myths and misconceptions? What are they if any?

Please describe the IUD offered, The constituent and its efficacy

Line 230. Please state the country origin and the company producing STATA.

Line 247: Fischer's exact test is not used for skewed data. It is used in place of Chi square if the expected frequencies in the cells < 5. (please confirm its use)

What non-parametric equivalent was utilized in place of Student's ttest where appropriate? (Please note that "S" in Student's ttest is capital letter as it is a name of a person)

Line 249 - 250 : Authors stated "Bivariate analysis was done using modified Poisson regression due to the prevalence being greater than 15%" I am a bit confused about this sentence. Authors may explain further and cite appropriate references.

Line 251: Please change "multivariate" to "multivariable".

Please state the type of model building. stepwise forward? backward elimination or prior selections?

Line 257. The prevalence should be reported as a percentage and its 95% confidence interval. Please authors should state 2 different prevalences: Prevalence of IUD among all women assessed and prevalence of IUD acceptors among those who accepted any form of contraceptives. (with their 95% confidence interval)

line 260 - 261: Authors stated: "Interaction between the study variables was assessed by forming two-way interaction terms and comparing the models using the likelihood ratio test" Which variables were utilized

for the interaction tests? why and why not? Is the selection based on literature of knowledge of the field? if based on the literature, please provide evidence/citations

Authors may need to state if the analysis was two tailed or one-tailed

Line 269- 273. Authors stated "Between 1st February 2023 and 30th September 2023, we screened 1,911 potential participants who received medical management of first trimester incomplete abortions and were willing to take up post abortion intrauterine contraception at the five health facilities. Out of these 271 potential participants, we enrolled 650 participants. The proportion of women who took up 272 IUDs after medical management of first trimester abortion, was 370/650 (56.9%)"

This is not clear. Please authors may need to draw a flow diagram to aid understanding. How was the 650 participants recruited from 1,911? Do you mean 1911 participants were ready to take any contraceptive or they were ready to take IUD? Becuase you now had a proportion 370/650? Who are the denominator here? Please add 95%CI to the prevalence of IUD uptake

Line 273: Majority generally means percentage of >70%. and not >50%

line 280: authors stated "A high proportion of the participants (56.6 %)" 56.6% is "a little above half" and not "high proportion"

please show the mean age in the Table 1

line 283 -284: authors stated ".......and had 2-3 children 283 (38.5%)." 38.5% is not "majority". it is about "one-third"

Table 1 should a column for those that accepted IUD, another column for those that did not take IUD , another column for those that did not accept any contraceptive and then the total.

For each variable, you may use appropriate statistical tool to compare the variables across the categories of studied groups

Table 1. Please report the mean or median age (as the case may be) in the Table

Table 1: Add year to age i.e "age (years)

Table 1. Change "below 20 years" to "<20"

Table 1. Change "Greater than 35years" to ">35"

Table 1: "Number of living children" is a discrete variable and may not be normally distributed. Authors should check for normality. If not normally distributed median, IQR should be reported instead of mean, SD.

Table 1. SD: Define SD as footnote of the table. Add appropriate symbol in the Table. See PLOS one website on formatting tables

Table 1. Some variables in Table 1 are not sociodemographic characteristics. Some are obstetrics or gynaecological characteristics. Sot the title may be "sociodemographic and gynaecological characteristics"

Table 1: HIV status is not part of the sociodemographic characteristics

Table 1 . please expunge "Current uptake of Post abortion contraceptive method(N=650) " from Table 1. You may even produce appropriate chart/figure with this.

Table 2. "Unadjusted PR (CI 95%)" shifted from where it should be in the Table. please correct it

Table 2. "NB". What is "NB" in the footnote? I will advise you remove it

Discussion: You should briefly restate your aim at the beginning of your discussion.

Line 292 - 295: Authors stated "At bivariate analysis; age, religion, current number of children, occupation, monthly income, 292 distance from health facility, spousal approval of the contraception, and co-habiting status, 293 were statistically associated with the use post abortion IUDs after the first trimester of 294 incomplete abortion with a p-value less than 0.2.(Table 2) " P-value pf 0.2 does not suggest Statistical significance. What is means are that they were variables that were utilized for the multivariable analysis because their P-values were < 0.2. Not because they are significant. Authors should rework this sentence

Libe 297: change "multivariate" to "multivariable".

So this model building is with a binary variable of (IUD uptake Vs no IUD uptake)??? this may mean that "no IUD uptake" is either uptake of other contraceptives or non-use of contraceptives??? Please clarify. The outcome of the model should be a binary variable of either (IUD uptake Vs no contraceptive uptake) or (IUD uptake Vs other contraceptive uptake). Your interpretation of the results should clearly incorporate your outcome out of the two scenarios stated above

Line 430: change "couldn’t " to could not. avoid contracted words in scientific writings

line 479: Availability of data and Materials: Authors may attach as supplementary materials, the anonymized minimal data underlying this study. It may be in excel, or stata format. (Except there are strong ethical and legal reasons against it)

Reviewers' comments:

Reviewer's Responses to Questions

Comments to the Author

1. Is the manuscript technically sound, and do the data support the conclusions?

Reviewer #1: Yes

Reviewer #2: Yes

Reviewer #3: Yes

Reviewer #4: Yes

2. Has the statistical analysis been performed appropriately and rigorously?

Reviewer #1: Yes

Reviewer #2: No

Reviewer #3: Yes

Reviewer #4: Yes

3. Have the authors made all data underlying the findings in their manuscript fully available?

Reviewer #1: Yes

Reviewer #2: Yes

Reviewer #3: Yes

Reviewer #4: Yes

4. Is the manuscript presented in an intelligible fashion and written in standard English?

Reviewer #1: Yes

Reviewer #2: Yes

Reviewer #3: Yes

Reviewer #4: Yes

5. Review Comments to the Author

Reviewer #1: REVIEW OF ARTICLE TITLED Uptake of Intrauterine Contraception after Medical Management of First Trimester Incomplete Abortion: A Cross-sectional study in central Uganda

Thank you for asking me to review the above titled article. My comments are as follows

• The article is well titled with an appreciation of its content and the methodology expected.

• The authors have chosen a contemporary topic aiming to present components of post abortion care as important strategies to prevent morbidity and mortality from unsafe abortion and its sequelae. This should be applauded

• The abstract is well written giving an overview of the aim of the study its methodology results and conclusion

• The introduction presents the background and basis for undertaking the study and by the end the reader appreciates the justification for this research paper

• The study design is appropriate for research of this nature

• There is no segregation of the index abortions into spontaneous or induced in this write up , and if induced what were the reasons for initiating the termination of pregnancy. This may have a bearing on choosing to have contraception after post abortion care or not

• How did the patients arrive at the decision to undergo medical management? Did they request for it ab initio or were they counselled on all options and allowed to choose? Its not clearly indicated here.

• Also who did the counselling for options of therapy? Was it done by those providing the definitive care ? and were they part of the staff trained for this study?

• Just out of curiosity, of the 1911 women recruited you enrolled 650 which was your sample size, but why the drop off of over 65% ? were there any specific trends noted or was it just the exclusion criteria?

• The presentation of results are acceptable and they derive from the aim and purported objectives of this research

• The discussion is well detailed and easy to read, they are based on the results presented with logical arguments and submissions

• The conclusion is acceptable

The study is an interesting one and shines a light on a topic that is usually taboo in Africa. The authors have presented a detailed rendition of their work .

I believe if the queries raised are answered appropriately it can be considered for publication

Thank you

Reviewer #2: Most of unwanted pregnancies end up in induced abortion ......68

Unnecessary references 5......70

Statement with ref 7, check and revise......73

Using 7 references for a statement seem overboard, 2-3 max would be advised....90,91

The content of description of the study setting is too much. Summary and make it more concise especially the management p

---

## [Decision Letter · Decision Letter 2]

9 Jun 2025

Dear Dr. Kayiga,

Thank you for submitting your manuscript to PLOS ONE. After careful consideration, we feel that it has merit but does not fully meet PLOS ONE’s publication criteria as it currently stands. Therefore, we invite you to submit a revised version of the manuscript that addresses the points raised during the review process.

We look forward to receiving your revised manuscript.

Kind regards,

Gbenga Olorunfemi, MBBS,MSC,FMCOG,FWASC,FWACOG

Academic Editor

PLOS ONE

Journal Requirements:

**Additional Editor Comments:**

line 134; Resmove "see" before "appendix"

Line 282 add "P-value = 0.57"

line 284. Insert a reference for this sentence/assertion

line 296 . Remove "see" from "see Fig". Do the same for all tables and fig

Table 2. Please include the Exact P-values of the adjusted PR in the Table.

Line 526 . Change I'm to "I am"

Please review all the references and ensure they conform with Vancouver referencing styles. Many of the references are not in line with Vancouver.

Reviewers' comments:

Reviewer's Responses to Questions

**Comments to the Author**

Reviewer #2: (No Response)

Reviewer #4: All comments have been addressed

2. Is the manuscript technically sound, and do the data support the conclusions?

Reviewer #2: Partly

Reviewer #4: Yes

3. Has the statistical analysis been performed appropriately and rigorously?

Reviewer #2: No

Reviewer #4: Yes

4. Have the authors made all data underlying the findings in their manuscript fully available?

Reviewer #2: No

Reviewer #4: (No Response)

5. Is the manuscript presented in an intelligible fashion and written in standard English?

Reviewer #2: Yes

Reviewer #4: Yes

Reviewer #2: The authors are yet to address some of the critical issues raised in the sample size calculation as 2 different sample size were used. Also there are some irregularities observed in the analysis for examples in table 1, the authors did not specify the test of association that was used but only inputted the P value.

Reviewer #4: (No Response)

**Do you want your identity to be public for this peer review?** For information about this choice, including consent withdrawal, please see our Privacy Policy

Reviewer #2: No

Reviewer #4: No

---

## [Author Response · Author response to Decision Letter 3]

14 Jun 2025

COLLEGE OF HEALTH SCIENCES

SCHOOL OF MEDICINE

DEPARTMENT OF OBSTETRICS AND GYNAECOLOGY

13th June 2025

RESPONSE TO REVIEWS COMMENTS:

Object: PONE-D-24-39454R2 “UPTAKE OF INTRAUTERINE CONTRACEPTION AFTER MEDICAL MANAGEMENT OF FIRST TRIMESTER INCOMPLETE ABORTION: A CROSS-SECTIONAL STUDY IN CENTRAL UGANDA”

With great pleasure, I’m thankful for your comments towards our manuscript. In response to the reviewers’ comments sent to us on 10th June 2025, we have revised the manuscript accordingly.

Comment Response to Comment Page No. and Line

JOURNAL REQUIREMENTS:

Additional Editor Comments:

line 134; Resmove "see" before "appendix"

Line 282 add "P-value = 0.57"

line 284. Insert a reference for this sentence/assertion

line 296 . Remove "see" from "see Fig". Do the same for all tables and fig

Table 2. Please include the Exact P-values of the adjusted PR in the Table.

Line 526 . Change I'm to "I am"

Please review all the references and ensure they conform with Vancouver referencing styles. Many of the references are not in line with Vancouver.

Reviewers' comments:

Reviewer's Responses to Questions

Comments to the Author

1. If the authors have adequately addressed your comments raised in a previous round of review and you feel that this manuscript is now acceptable for publication, you may indicate that here to bypass the “Comments to the Author” section, enter your conflict of interest statement in the “Confidential to Editor” section, and submit your "Accept" recommendation.

Reviewer #2: (No Response)

Reviewer #4: All comments have been addressed

2. Is the manuscript technically sound, and do the data support the conclusions?

Reviewer #2: Partly

Reviewer #4: Yes

3. Has the statistical analysis been performed appropriately and rigorously?

Reviewer #2: No

Reviewer #4: Yes

4. Have the authors made all data underlying the findings in their manuscript fully available?

Reviewer #2: No

Reviewer #4: (No Response)

5. Is the manuscript presented in an intelligible fashion and written in standard English?

Reviewer #2: Yes

Reviewer #4: Yes

6. Review Comments to the Author

Reviewer #2: The authors are yet to address some of the critical issues raised in the sample size calculation as 2 different sample size were used. Also there are some irregularities observed in the analysis for examples in table 1, the authors did not specify the test of association that was used but only inputted the P value.

Which sampling technique did you used in this study?

Multi-satge technique

Purposive technique at the point of participant recruitment

How relevant is this paragraph to this study and your sample size of 650?

Analysis stated here and did not reflect on the tables in the result section should be deleted.

Kindly re-write this section in a simple and clearer format

Add the name of this flow chart to it eg PRISMA

Which of the test statistics did you do here. Only P-value and no test statistics

Kindly state whether a test of normality was done before Median +/- interquartile range was done

Reviewer #4: (No Response)

The reference list has been reviewed and corrected as advised.

The manuscript has been revised accordingly to remove “see” before “appendix”.

P-value = 0.57 has been added as advised.

A reference (Hagiwara, Y. and Y. Matsuyama, Goodness-of-fit tests for modified poisson regression possibly producing fitted values exceeding one in binary outcome analysis. Statistical Methods in Medical Research, 2024. 33(7): p. 1185-1196.) has been added as recommended.

The manuscript has been revised accordingly to remove “see” before all tables and fig. P-values of the adjusted PR have been added in Table 2.

The write up has been edited as advised.

The references have been revised to align with the Vancouver referencing style.

Thanks so much for the feedback.

We appreciate the feedback. We have revised the manuscript extensively.

We appreciate the feedback. We have revised the statistical analysis comprehensively.

For ethical reasons to ensure patient privacy, only anonomyous data presented at group level,

are accessible from the corresponding author on request.

Thanks so much for this feedback.

The critical issues as raised by the reviewer on sample size have been comprehensively addressed in the revised manuscript. The test of association as used in Table 1, have been added as advised.

The manuscript has been revised. The write up now appears as “We used purposive sampling where participants who met the eligibility criteria and interested in participating in the study, were enrolled consecutively and given their preferred contraceptive choices, until the desired sample size was achieved”.

We had two objectives in this study;

1. To determine the prevalence of uptake post abortion intrauterine contraception after medical management of first trimester incomplete abortion.

2. To determine the factors associated with the uptake of post abortion intrauterine contraception after medical management of first trimester incomplete abortion.

The relevance of the paragraph is to show the assumptions we used to calculate the sample size for the first objective on the prevalence of uptake of the post abortion intrauterine contraception. We used a study “Benson J, Andersen K, Healy J, Brahmi D. What factors contribute to postabortion contraceptive uptake by young women? A program evaluation in 10 countries in Asia and sub-Saharan Africa. Global Health: Science and Practice, 2017. 5(4): p. 644-657”. In this study the prevalence of post abortion IUDs in ten countries in Asia ans Africa, was 11%. Using Open Epi, we got a sample size of 190 participants.

To determine sample size for the second objective on factors associated with the uptake of post abortion intrauterine contraception, we used a study “Makenzius M, Faxelid E, Gemzell-Danielsson K, Odero TM, Klingberg-Allvin M, Oguttu M. Contraceptive uptake in post abortion care—Secondary outcomes from a randomised controlled trial, Kisumu, Kenya. PloS one, 2018. 13(8): p. e0201214”. One of the significant factors to the uptake, was the participants’ age. Using Open Epi, we needed a sample size of 642 participants. We compensated for a non-response and had a total sample size of 650 participants. As required, we chose the larger sample size of the two objectives and used this sample to achieve both objectives. It is to this end that we have the sample size of 650 in our study.

The relevance of all these paragraphs, is to enable replicability of study findings in other settings as required of all researchers in regards to the assumptions used to determine our sample size.

Test statistics have been added to the Table 1 as advised. The write up of the analysis section has been revised accordingly.

The name of the flow chart has been edited as advised. It now appears as “Figure 1: Modified PRISMA Flow diagram of Participants enrolled in the Uptake of Post abortion intrauterine contraception after medical management of first trimester incomplete abortion

The appropriate test statistics have been added to Table 1 as advised.

A test of normality was done before and when the data was not normally distributed, median and interquartile range were used. The write up now appears as “Univariate analysis was done using frequencies and percentages for categorical variables and median and interquartile ranges for continuous variables where appropriate. A test of normality was performed to determine whether data was normally distributed or not.

Page 22-27, line 551-758

Page 6, line 133

Page 10, line 283

Page 10, line 285

Page 14, line 364

Page 20, line 517

Page 22-27, line 551-758

Page 7, line 178-181

Page 7-8, line 185-198

Page 12, line 338

Page 11, line 298

Page 12, line 338

Page 9, line 243-245

Yours truly,

Dr Herbert Kayiga

Corresponding Author,

Department of Obstetrics and Gynaecology

Makerere University College of Health Science

+256777855063, hkayiga@gmail.com

---

## [Decision Letter · Decision Letter 3]

4 Sep 2025

Dear Dr. Kayiga,

Thank you for submitting your manuscript to PLOS ONE. After careful consideration, we feel that it has merit but does not fully meet PLOS ONE’s publication criteria as it currently stands. Therefore, we invite you to submit a revised version of the manuscript that addresses the points raised during the review process.

**ACADEMIC EDITOR: Please respond to all reviewers comments**

We look forward to receiving your revised manuscript.

Kind regards,

Ahmed Mohamed Maged, MD

Academic Editor

PLOS ONE

Journal Requirements:

Reviewers' comments:

Reviewer's Responses to Questions

**Comments to the Author**

Reviewer #2: All comments have been addressed

Reviewer #3: (No Response)

Reviewer #4: All comments have been addressed

Reviewer #5: (No Response)

Reviewer #6: All comments have been addressed

Reviewer #7: All comments have been addressed

2. Is the manuscript technically sound, and do the data support the conclusions?

Reviewer #2: Yes

Reviewer #3: Yes

Reviewer #4: Yes

Reviewer #5: No

Reviewer #6: Yes

Reviewer #7: Yes

3. Has the statistical analysis been performed appropriately and rigorously?

Reviewer #2: Yes

Reviewer #3: Yes

Reviewer #4: Yes

Reviewer #5: Yes

Reviewer #6: Yes

Reviewer #7: Yes

4. Have the authors made all data underlying the findings in their manuscript fully available?

Reviewer #2: Yes

Reviewer #3: Yes

Reviewer #4: Yes

Reviewer #5: Yes

Reviewer #6: Yes

Reviewer #7: (No Response)

5. Is the manuscript presented in an intelligible fashion and written in standard English?

Reviewer #2: Yes

Reviewer #3: Yes

Reviewer #4: Yes

Reviewer #5: Yes

Reviewer #6: Yes

Reviewer #7: Yes

Reviewer #2: (No Response)

Reviewer #3: I think the writers have done very well with their data analysis and discussion of their results. This is another brilliant article towards advancement of women’s health.

However, some of the concerns raised in the previous review were not addressed although majority of them have been addressed.

Below are some comments that I think need to be attended to as well. This is geared towards making the article better for all.

Reviewers comments

1. Line 98….. two percent among……. and 15% among …….,,

I suggest the writer change two percent to 2% or change the 15% to fifteen percent to ensure uniformity. The same must be done for line 311 and all others.

2. Line 141 and 148 are contradictory. In 141, the writers say ultrasound was one of the modalities for dating the pregnancy. But in line 148 they say they only used ultrasound when team suspected incomplete miscarriage or device expulsion. I think they must reconcile the two statements.

3. Lines 150-156 how did the writers obtain consent from the minors who participated in the study(less 18 years ). Did they involve their parents/guardians?

4. Lines 175- 181. should be revised . It’s difficult to understand what the writers are saying.

I think the sentence should read, …At the post abortion clinic, potential participants with incomplete abortion were given comprehensive information on either ……..

5. Lines 178-181 “ In Uganda , nurses-midwives are permitted to perform abortion…..” I do not see the relevance of this statement under data collection. It may be appropriate to insert it under introduction.

1. Lines 297-298 should be captured on the table of results

The writers may address the issues raised above.

Reviewer #4: My comments and concerns on the manuscript have been addressed in the first round of review by the authors.

Reviewer #5: This manuscript addresses an important public health issue—determinants of post-abortion intrauterine device (IUD) acceptance. While the topic is timely, the study does not provide sufficiently novel insights beyond what is already known from prior research in similar contexts.

1. The cross-sectional design and the exclusive inclusion of women who already expressed an interest in IUDs introduce severe selection bias. The resulting findings cannot be generalized to the broader population of post-abortion women.

2. No adjustment was made for clustering by facility. Given that counseling practices, provider attitudes, and service availability vary widely across facilities, the lack of multilevel or cluster-adjusted analysis undermines the reliability of the reported associations.

3. Important details of participant recruitment, exclusion, and missing data management are insufficiently described.

4. Safety outcomes and adverse events central to post-abortion contraceptive research are not being reported. Please specify.

5. “Women chose between the Copper IUDs (Nova T®, Bayer AG, Berlin, Germany) and the Levonorgestrel (LNG) IUDs (Mirena®, Bayer AG, Berlin, Germany).” How did these women make their choices? They're not knowledgeable on the subject anyway.

6. “1.9%) participants had exploratory laparotomies following ectopic pregnancies. (Fig 1)” Weren't only incomplete abortion participants included?

Reviewer #6: No further questions or concerns. at this time for the authors. All questions have been answered to the reviewer's satisfaction.

Reviewer #7: (No Response)

**Do you want your identity to be public for this peer review?** For information about this choice, including consent withdrawal, please see our Privacy Policy

Reviewer #2: No

Reviewer #3: No

Reviewer #4: No

Reviewer #5: No

Reviewer #6: No

Reviewer #7: No

---

## [Author Response · Author response to Decision Letter 4]

9 Sep 2025

COLLEGE OF HEALTH SCIENCES

SCHOOL OF MEDICINE

DEPARTMENT OF OBSTETRICS AND GYNAECOLOGY

09th September 2025

RESPONSE TO REVIEWS COMMENTS:

Object: PONE-D-24-39454R3 “UPTAKE OF INTRAUTERINE CONTRACEPTION AFTER MEDICAL MANAGEMENT OF FIRST TRIMESTER INCOMPLETE ABORTION: A CROSS-SECTIONAL STUDY IN CENTRAL UGANDA”

With great pleasure, I’m thankful for your comments towards our manuscript. In response to the reviewers’ comments sent to us on 4th September 2025, we have revised the manuscript accordingly.

Comment Response to Comment Page No. and Line

JOURNAL REQUIREMENTS:

Reviewers' comments:

Reviewer's Responses to Questions

Comments to the Author

1. If the authors have adequately addressed your comments raised in a previous round of review and you feel that this manuscript is now acceptable for publication, you may indicate that here to bypass the “Comments to the Author” section, enter your conflict of interest statement in the “Confidential to Editor” section, and submit your "Accept" recommendation.

Reviewer #2: All comments have been addressed

Reviewer #3: (No Response)

Reviewer #4: All comments have been addressed

Reviewer #5: (No Response)

Reviewer #6: All comments have been addressed

Reviewer #7: All comments have been addressed

2. Is the manuscript technically sound, and do the data support the conclusions?

Reviewer #2: Yes

Reviewer #3: Yes

Reviewer #4: Yes

Reviewer #5: No

Reviewer #6: Yes

Reviewer #7: Yes

3. Has the statistical analysis been performed appropriately and rigorously?

Reviewer #2: Yes

Reviewer #3: Yes

Reviewer #4: Yes

Reviewer #5: Yes

Reviewer #6: Yes

Reviewer #7: Yes

4. Have the authors made all data underlying the findings in their manuscript fully available?

Reviewer #2: Yes

Reviewer #3: Yes

Reviewer #4: Yes

Reviewer #5: Yes

Reviewer #6: Yes

Reviewer #7: (No Response)

5. Is the manuscript presented in an intelligible fashion and written in standard English?

Reviewer #2: Yes

Reviewer #3: Yes

Reviewer #4: Yes

Reviewer #5: Yes

Reviewer #6: Yes

Reviewer #7: Yes

6. Review Comments to the Author

Reviewer #2: (No Response)

Reviewer #3: I think the writers have done very well with their data analysis and discussion of their results. This is another brilliant article towards advancement of women’s health.

However, some of the concerns raised in the previous review were not addressed although majority of them have been addressed.

Below are some comments that I think need to be attended to as well. This is geared towards making the article better for all.

Reviewers comments

1. Line 98….. two percent among……. and 15% among …….,,

I suggest the writer change two percent to 2% or change the 15% to fifteen percent to ensure uniformity. The same must be done for line 311 and all others.

2. Line 141 and 148 are contradictory. In 141, the writers say ultrasound was one of the modalities for dating the pregnancy. But in line 148 they say they only used ultrasound when team suspected incomplete miscarriage or device expulsion. I think they must reconcile the two statements.

3. Lines 150-156 how did the writers obtain consent from the minors who participated in the study(less 18 years ). Did they involve their parents/guardians?

4. Lines 175- 181. should be revised . It’s difficult to understand what the writers are saying.

I think the sentence should read, …At the post abortion clinic, potential participants with incomplete abortion were given comprehensive information on either ……..

5. Lines 178-181 “ In Uganda , nurses-midwives are permitted to perform abortion…..” I do not see the relevance of this statement under data collection. It may be appropriate to insert it under introduction.

1. Lines 297-298 should be captured on the table of results

The writers may address the issues raised above.

Reviewer #4: My comments and concerns on the manuscript have been addressed in the first round of review by the authors.

Reviewer #5: This manuscript addresses an important public health issue—determinants of post-abortion intrauterine device (IUD) acceptance. While the topic is timely, the study does not provide sufficiently novel insights beyond what is already known from prior research in similar contexts.

1. The cross-sectional design and the exclusive inclusion of women who already expressed an interest in IUDs introduce severe selection bias. The resulting findings cannot be generalized to the broader population of post-abortion women.

2. No adjustment was made for clustering by facility. Given that counseling practices, provider attitudes, and service availability vary widely across facilities, the lack of multilevel or cluster-adjusted analysis undermines the reliability of the reported associations.

3. Important details of participant recruitment, exclusion, and missing data management are insufficiently described.

4. Safety outcomes and adverse events central to post-abortion contraceptive research are not being reported. Please specify.

5. “Women chose between the Copper IUDs (Nova T®, Bayer AG, Berlin, Germany) and the Levonorgestrel (LNG) IUDs (Mirena®, Bayer AG, Berlin, Germany).” How did these women make their choices? They're not knowledgeable on the subject anyway.

6. “1.9%) participants had exploratory laparotomies following ectopic pregnancies. (Fig 1)” Weren't only incomplete abortion participants included?

Reviewer #6: No further questions or concerns. at this time for the authors. All questions have been answered to the reviewer's satisfaction.

Reviewer #7: (No Response)

7. PLOS authors have the option to publish the peer review history of their article (what does this mean?). If published, this will include your full peer review and any attached files.

Do you want your identity to be public for this peer review? For information about this choice, including consent withdrawal, please see our Privacy Policy.

Reviewer #2: No

Reviewer #3: No

Reviewer #4: No

Reviewer #5: No

Reviewer #6: No

Reviewer #7: No

This information is well received.

Thanks so much for this feedback.

Thanks so much for the feedback.

Thanks so much for the feedback.

Thanks for the feedback.

Thanks so much for this feedback.

The manuscript has been revised as advised. The write up now appears as “Although, intrauterine devices (IUDs) are readily available in Uganda, their user rate has stalled at 2% among the currently married and 15% among the sexually active unmarried women.”

Of the other available contraceptive methods, 121(18.6%) women opted for injectable Depo-Provera (DMPA), 116(17.9%) women opted for implants, 35(5.4%) women opted for oral contraceptive pills, 5 (0.8%) of the women opted for condoms, and 3 (0.5%) women opted for periodic abstinence.

A third of the participants were aged between (25-29) years. Over 50% of the participants who had abortions, had had a livebirth before. The majority of the participants who had had abortions were living with a partner (84.0%). Forty percent of the participants who had had abortions, had 2-3 children. Only (7)1% of the participants had a monthly income greater than Ugx shillings 1,000,000/= (USD 270). (Table 1)”

Thanks so much for this observation. We have deleted the aspect of using ultrasound scan in dating of the gestational age. The write up of this section now appears as “The criteria for identifying study participants with incomplete abortion was based on experience of any of the following conditions: a confirmation of pregnancy by any of the following methods; a positive urine HCG, or calculation from the first day of the last menstrual period. Furthermore, history of lower abdominal pain, and vaginal bleeding, before 12 weeks of gestation was needed to confirm occurrence of first trimester incomplete abortion. Clinical evaluation that included ascertaining cervical dilatation, feeling or visualization of products of conception on vaginal or speculum examination by the clinical team, was conducted to confirm the diagnosis of first trimester incomplete abortion. As indicated in evidence(30), ultrasonography in our study was only used when the clinical team was suspecting incomplete abortion or IUD expulsions.”

In Uganda minors who conceive are referred to as “emancipated minors” and are allowed to give consent according to the ethical guidelines. In this regard, consent was obtained from minors 15 years or above. Parents/guardians were not involved in the research as permited by the attached ethical approvals from the School of Medicine and Ethics review Committee and the Uganda National Council for Science and Technology.

We appreciate the caution. The intention was to clarify that the participants had to be within 4 weeks after the abortion care. We have however deleted the other aspects. The revised manuscript now appears as “At the post abortion follow-up visit, potential participants were given comprehensive information on either surgical evacuation (uterine aspiration) or medical evacuation with sublingual misoprostol 400 mcg by trained nurse-midwives.”

The write up has been deleted from the data collection procedure.

The details as indicated in Lines 297-298 are now reflected in the fig 1 of the revised manuscript.

Thanks so much for this feedback.

We appreciate the feedback.

This study informs policy that even when women express interest in using post abortion intrauterine contraception and all the materials required are fully paid off, nearly 40% of them will opt not to use the method. Though the reviewer might think otherwise, there was a need to explore the factors that influence the uptake of post abortion intrauterine contraception. Since all potential participants were given all the information about all the available contraceptive options, we minimized social desirability bias. With our large sample size of 650 participants from rural, peri urban and urban areas, we are convinced that our findings can be generalized to other areas with similar contextual settings. Our study suggests that further inquiry should be underway to determine the barriers and facilitators of the uptake of post abortion IUDs to minimize on the subsequent unintended pregnancies after first trimester abortions.

Since the study sites had similar context setting, health cadres, all being public facilities offering free family planning services in central Uganda, the relevance of clustering was watered down and the IRB approved an amendment that excluded the clustering effect. The amendment has been attached to this effect.

Though the wording of the subsections might differ, we are convinced that the participant recruitment details are covered under “the participant recruitment section” with further details under “Data collection procedure.” We highlight which patients were identified as potential participants, and the desired patient parameters we sought for. The inclusion and exclusion criteria are indicated. We give an elaborate detail on who the research team members were for the study. We are happy to offer more details as we shall be advised in this regard by the reviewer.

This was a cross sectional study which might have challenges with causal inference to establish cause-and-effect relationships but in the primary study all these aspects are mentioned. We have edited the revised manuscript to add further details of the primary study and the safety and adverse events.

The write up now appears as “We used a cross-sectional study to determine the level and factors associated with uptake of post abortion IUDs following medical management of first trimester incomplete abortions at five public facilities between 1st February 2023 and 30th September 2023. The participants selected were part of those who were enrolled in the primary study that was non-inferiority open-label randomized controlled study, that compared the expulsion and continuation rates following early insertion (within one week) versus standard insertion (2-4 weeks) after medical management of first trimester incomplete abortion(28). The trial was registered at ClinicalTrials.gov NCT05343546.”

Both copper and levonorgestrel IUDs are readily accessible in Uganda in nearly 90% of the health facilities. It is to this end that we assumed that the women were knowledgeable of the IUD choices. This was also reinforced by the comprehensive contraceptive counselling services that was offered by the research team prior to enrolment or even utilizing the method.

As requested by one of the reviewers before, a schematic flow of the participants was requested for participants that were included and excluded from the study as indicated in Figure 1. Since the participants had earlier on reported par vaginal bleeding, and lower abdominal pain, though mentioned in text, they were excluded from the study.

Thanks so much for the feedback.

Figure 1 has been

---

## [Decision Letter · Decision Letter 4]

24 Sep 2025

Uptake of Intrauterine Contraception after Medical Management of First Trimester Incomplete Abortion: A Cross-sectional study in central Uganda

PONE-D-24-39454R4

Dear Dr. Kayiga,

We’re pleased to inform you that your manuscript has been judged scientifically suitable for publication and will be formally accepted for publication once it meets all outstanding technical requirements.

Kind regards,

Ahmed Mohamed Maged, MD

Academic Editor

PLOS ONE

Additional Editor Comments (optional):

Reviewer #3:

Reviewers' comments:

Reviewer's Responses to Questions

**Comments to the Author**

Reviewer #3: All comments have been addressed

2. Is the manuscript technically sound, and do the data support the conclusions?

Reviewer #3: Yes

3. Has the statistical analysis been performed appropriately and rigorously?

Reviewer #3: Yes

4. Have the authors made all data underlying the findings in their manuscript fully available?

Reviewer #3: Yes

5. Is the manuscript presented in an intelligible fashion and written in standard English?

Reviewer #3: Yes

Reviewer #3: (No Response)

**Do you want your identity to be public for this peer review?** For information about this choice, including consent withdrawal, please see our Privacy Policy

Reviewer #3: No

---

## [Editor Report · Acceptance letter]

PONE-D-24-39454R4

PLOS ONE

Dear Dr. Kayiga,

I'm pleased to inform you that your manuscript has been deemed suitable for publication in PLOS ONE. Congratulations! Your manuscript is now being handed over to our production team.

Kind regards,

on behalf of

Professor Ahmed Mohamed Maged

Academic Editor

PLOS ONE